# Freezing and Thawing Resistance of Fine Recycled Concrete Aggregate (FRCA) Mixtures Designed with Distinct Techniques

**DOI:** 10.3390/ma15041342

**Published:** 2022-02-11

**Authors:** Cassandra Trottier, Mayra T. de Grazia, Hian F. Macedo, Leandro F. M. Sanchez, Gabriella P. de Andrade, Diego J. de Souza, Olga Naboka, Gholamreza Fathifazl, Pierre-Claver Nkinamubanzi, André Demers

**Affiliations:** 1Department of Civil Engineering, Faculty of Engineering, University of Ottawa, Ottawa, ON K1N 6N5, Canada; mayra.grazia@uottawa.ca (M.T.d.G.); hdefr095@uottawa.ca (H.F.M.); gandr070@uottawa.ca (G.P.d.A.); dsouz025@uottawa.ca (D.J.d.S.); 2National Research Council Canada (NRCC), Ottawa, ON K1A 0R6, Canada; olga.naboka@nrc-cnrc.gc.ca (O.N.); gholam-reza.fathi-fazl@nrc-cnrc.gc.ca (G.F.); pierre-claver.nkinamubanzi@nrc-cnrc.gc.ca (P.-C.N.); 3CANMET Energy (NRCAN), Ottawa, ON K1V 1E1, Canada; andre.demers@canada.ca

**Keywords:** fine recycled concrete aggregates, low cement concrete, eco-friendly concrete, the durability of FRCA concrete, mix design, particle packing model, freezing and thawing, microscopic assessment

## Abstract

The pressure to use sustainable materials and adopt practices reducing the carbon footprint of the construction industry has risen. Such materials include recycled concrete aggregates (RCA) made from waste concrete. However, concrete made with RCA often presents poor fresh and hardened properties along with a decrease in its durability performance, especially when using its fine fraction (i.e., FRCA). Most studies involving FRCA use direct replacement methods (DRM) to proportion concrete although other techniques are available such as the Equivalent Volume (EV) and Particle Packing Models (PPMs); yet their impact on the durability performance, especially its performance against freezing and thawing (F/T), remains unknown. This work, therefore, appraises the F/T resistance of FRCA mixtures proportioned through various mix proportioning techniques (i.e., DRM, EV and PPMs), produced with distinct crushing processes (i.e., crusher’s fines vs. finely ground). The results show that the mix design technique has a significant influence on the FRCA mixture’s F/T resistance where PPM-proportioned mixtures demonstrate the best overall performance, exceeding the specified requirements while DRM-proportioned mixtures failed F/T resistance requirements. Moreover, the crushing process plays an important role in the recycled mixtures’ cracking behavior under F/T cycles, where less processing leads to fewer cracks while remaining the most sustainable option overall.

## 1. Introduction

Tackling climate change is among the top priorities worldwide, with emphasis on reducing CO_2_ emissions. Canada has set its target of reaching the so-called net-zero by 2050 [1]. There is indeed an urgency to adopt alternative solutions to reduce the carbon footprint in the construction industry. The production of Portland cement alone amounts to approximately 5% of global CO_2_ emissions [2]. Moreover, sources of the non-renewable natural aggregates are becoming depleted nearby major continuously expanding urban centers resulting in longer transportation distances and, thus, an increased carbon footprint [3,4,5]. Therefore, the use of recycled concrete aggregates (RCA) has been progressively considered as a sustainable alternative to the non-renewable natural aggregate. RCA can help decrease the carbon footprint of new concrete (i.e., lessening the amount of new Portland cement required when accounting for the residual cement paste into the mixture), limiting the use of natural resources and reducing landfill’s usage, provided that the RCA is in closer proximity to the concrete plant/construction site [6,7,8,9].

RCA is produced by crushing returned or demolished concrete into aggregate-sized fractions. Generally, the coarse RCA fraction is used in new concrete construction, often limited to non-structural applications due to concerns over the material’s variability, quality and presence of impurities (i.e., other waste debris), whereas the fine RCA (FRCA) fraction is rejected, being considered as a by-product or waste [4,5]. 

Much of the research on RCA implicitly treats the material as being homogeneous and directly replaces certain proportions of natural aggregate by RCA through partial or full direct replacement methods (DRM) in the concrete mixture without further consideration of its multi-phase nature, especially for FRCA. Indeed, the multi-phase nature of FRCA, consisting of residual cement paste (RCP) adhered to residual sand (RS) and/or fractured coarse original virgin aggregate (OVA), influences the performance of recycled concrete mixtures, often resulting in inferior mechanical and durability properties. Moreover, given the high variability of RCA, accurately estimating and/or predicting the fresh and hardened state performance of recycled mixtures designed through the DRM method remains a challenge [10,11,12,13]. However, recent studies demonstrated that when the unique microstructure of the RCA is accounted for in the mix proportioning of recycled mixtures [7,8,9,14], adequate fresh and hardened state behavior might be obtained. Amongst the mix design techniques successfully implemented in this regard, the Equivalent Volume (EV) [9] method showed to be quite promising, enabling a replacement ratio of 100% while achieving the targeted properties. On the other hand, several advanced mixture proportioning techniques based on particle packing models (PPMs) were used for both coarse RCA [15,16] and FRCA [17,18] mixtures, yielding interesting results, especially in the hardened state (i.e., compressive strength). Nevertheless, the impact of using distinct mix design techniques on the durability and long-term performance of recycled mixtures incorporating FRCA is yet to be determined. This work appraises the durability performance against freeze and thawing (F/T) cycles of recycled concrete mixtures designed through various proportioning techniques (i.e., DRM, EV and PPM) and crushing techniques while incorporating 100% of FRCA.

## 2. Background

### 2.1. Multi-Phase Nature of Recycled Concrete Aggregates (RCA)

Concrete waste is crushed then sieved to obtain certain particle sizes. Fine recycled concrete aggregates (FRCA) are normally considered as a by-product of coarse RCA (CRCA) production. Consequently, CRCA is composed of mainly original virgin aggregate (OVA) and residual mortar (RM); the latter can even represent up to 60% of the CRCA volume [19]. Meanwhile, FRCA is generally composed of RM, which can be further divided into residual cement paste (RCP) and residual sand (RS). Some fractured OVA may detach during the crushing process and thus can be part of the FRCA particles. Therefore, FRCA can be composed of only RM, only OVA or RS and a combination of RM and OVA as highlighted in Figure 1 through the yellow, green and blue contours, respectively. In addition, FRCA contains original interfacial transition zones (ITZ) between the OVA/RS and RCP, enhancing the multi-phase nature of the aggregate. 

Physical properties of the FRCA particles are influenced by several factors such as the amount of RCP found in FRCA particles that can be influenced by the type and quality of the OVA and RS (i.e., lithotype, texture, shape, etc.), the RCP’s quality (i.e., porosity and mechanical properties) and the FRCA crushing process (i.e., crusher type, crushing series and energy, etc.), leading to important variability among the final product [20,21]. Moreover, when RCA is used in concrete, new ITZs are formed between the OVA/RS to new cement paste and residual to new cement paste interfaces; thus, adding another component to the distinct microstructure of concrete made of FRCA.

FRCA is generally considered a very low-quality material due to its high amount of RM, resulting in low specific gravity, high porosity and absorption, the latter of which corresponds to the increased water demand of the FRCA concrete. Some studies were performed to quantify the RM of FRCA [22,23,24]; however, the quantification of the RCP is more relevant as FRCA may be composed of only RM, which makes the results obtained on those studies somewhat inconsistent [25,26]. De Andrade et al. [18] used a discrete PPM approach, the so-called Compressible Packing Model (CPM) developed by De Larrard [27], to proportion recycled concrete mixtures made of 100% FRCA and found that better control over the properties of the material might be achieved despite the FRCA features; however, the RCP in the FRCA was quantified in this study only for comparative purposes. 

Despite all recent advancements in recycled concrete using FRCA, when compared to conventional concrete (CC), recycled concrete mixtures designed through conventional methods using partial (over 30%) or full FRCA replacement were observed to often yield lower performance in the fresh (i.e., high consistency and low flowability) and hardened (i.e., low compressive strength and stiffness) states along with durability [11,28,29,30,31,32,33,34,35,36,37,38,39,40,41]. 

### 2.2. Available Mixture Proportioning Techniques for RCA Concrete 

#### 2.2.1. Direct Replacement Method (DRM)

Direct replacement methods (DRM), either partial/full or by weight/volume, are one of the earliest attempts to mixture proportion concrete using FRCA. Studies show that a replacement ratio of less than 30% provides optimal results [29,31,32,33,38,42,43,44], although suitable properties were achieved using 100% replacement in some cases. Moreover, results of recycled concrete mixtures designed through DRM were verified to be extremely variable. As such, it was suggested by Nedeljković et al. [45] in a critical review of the use of FRCA to develop new approaches when proportioning recycled mixtures.

#### 2.2.2. Equivalent Volume (EV)

The Equivalent Volume (EV) [9] is the most recent mix design technique created that accounts for RCP content adhered to the CRCA particles to proportion concrete made of CRCA, allowing up to 100% of CRCA replacement while achieving the targeted properties. This method was developed based on the Equivalent Mortar Volume (EMV [7]) and EMV modified (EMV-mod [8]), overcoming the issues presented in the fresh state and eco-efficiency, respectively. The EV mix design technique develops a CRCA concrete that bears the same total volume of cement paste (i.e., residual and new) and the same total volume of aggregates (i.e., OVA and residual and new sand) as its companion CC. This technique was used and adapted for FRCA in a feasibility study [46].

#### 2.2.3. Particle Packing Models (PPMs)

Particle packing models (PPMs) are advanced mix design techniques used to improve the performance of concrete mixtures in both fresh and hardened states while optimizing the granular system, resulting in an improved packing density and reduced porosity [47,48,49,50,51]. Hence, concrete mix proportioned through PPMs often present lower Portland cement content and enhanced durability aspects when compared to CC. Although PPMs and other similar techniques are rarely used for RCA, previous studies [16,17,18] showed promising results. PPMs are divided into discrete and continuous models, the former of which assumes the existence of a given number of discrete particle sizes that are rearranged to reach the maximum packing density [16,49,51], whereas the latter assumes a continued distribution of particles within the system [49,51]. The Modified Andreasen model (also known as the Alfred model) is the most recent continuous PPM, which calculates the optimum particle size distribution based on a coefficient of distribution (*q*) along with the largest (*D_L_*) and smallest (*D_S_*) particle diameter present in the system. The coefficient of distribution is normally adopted based on the fresh state requirements of the mixture. Distribution coefficients ranging between 0.20 and 0.23 are often selected for self-consolidating mixtures, while values between 0.26 and 0.28 are normally targeted for vibrated and/or pumped concrete. It is worth noting that 0.37 is the coefficient of distribution that yields the highest packing density and thus lowest porosity to the granular system as per [51]. The optimum particle size distribution is calculated as per Equation (1):(1)$$CPFT=100\times \left(\frac{D_{P}{}^{q}-D_{S}{}^{q}}{D_{L}{}^{q}-D_{S}{}^{q}}\right),$$
where *CPFT* is the cumulative (volume) percent finer than *D_P_* and *D_P_* is the particle diameter.

### 2.3. Durability and Long-Term Performance of FRCA Concrete

#### 2.3.1. Non-Destructive Tests (NDT) and Stiffness Damage Test (SDT)

A convenient evaluation of the inner quality of concrete mixtures may be performed using non-destructive techniques (NDT), such as ultrasonic pulse velocity (UPV) and surface electrical resistivity (ER). Such techniques were verified to be suitable to appraise the inner quality of CC mixtures subjected to aggressive/harsh environments [52] along with the effects of RCA replacement [9,42,53,54,55]. Studies show that the higher the RCA replacement proportioned with DRM, the lower the NDT results due to the lower inner quality of the RCA [42,56,57].

Moreover, regardless of the mixture proportioning technique used, RCA concrete mixtures generally present lower stiffness when compared to CC [39,58,59], likely associated with the presence of microcracks within the RCA. The Stiffness Damage Test (SDT-[60,61,62]) was shown to be a very reliable tool to quantify physical damage in concrete through the stiffness damage index (SDI) and plastic deformation index (PDI), which relate the number of inner cracks with either the energy dissipated over the loading–unloading cycles or the unrecoverable deformation after five loading cycles, respectively. Moreover, the modulus of elasticity of the material can be calculated using the average of the secant modulus of the second and third cycles [61,62] through the SDT approach. However, the SDT has never been used to evaluate the inner quality and flaws (i.e., microcracks) of FRCA concrete. 

#### 2.3.2. Freeze-Thaw Resistance

Freeze-thaw (F/T) resistance of CC is required in Canada’s harsh environments and is linked to various factors such as the aggregate features, cement paste porosity and free water availability in the system. These parameters may be even more critical for recycled concrete since its higher porosity may contribute to an increase in the material’s absorption and permeability [63,64]. Some studies have shown promising results regarding the F/T resistance of concrete made with FRCA [29,30,54]. However, the effect of highly porous aggregates such as RCA on permeability is not yet fully understood as it may play a dual role: highly porous aggregates are normally weaker and easily saturated, whereas increased permeability leads to dissipation of internal hydraulic pressure that may improve F/T resistance [65]. 

#### 2.3.3. The Damage Rating Index (DRI)

The Damage Rating Index (DRI) is a microscopic tool developed to assess the damage and its extent in CC [66,67,68,69,70]. Using a stereomicroscope at 16× magnification, distress features (i.e., cracks) are counted in 1 cm × 1 cm squares drawn on the surface of a polished concrete section after which weighting factors [71] are applied to those features to balance their relative importance towards the distress mechanism and normalized to 100 cm^2^ for comparative purposes. Moreover, the extended version of the DRI (i.e., without applying weighting factors) can be used to evaluate the crack propagation and distribution within a CC [66,68] as well as recycled concrete [72,73,74].

Nevertheless, the durability performance of concrete made of FRCA, namely its freezing and thawing resistance, while using several distinct mixture proportioning techniques (i.e., DRM, EV and PPMs) and crushing processes (i.e., crushing fines vs. fully ground) remains mostly unknown. Moreover, techniques such as the SDT and DRI are novel test procedures used to evaluate the inner quality and or damage degree of concrete and have never been used to evaluate concrete made of FRCA before.

## 3. Scope of the Work

Currently, there is a lack of studies on the durability and long-term performance of RCA, especially FRCA, which hinders its use in the industry. Moreover, the impact of distinct mix design procedures on the durability performance of recycled concrete mixtures made of FRCA is mostly unknown. Thus, this work aims to evaluate the durability related properties, namely the resistance against freezing and thawing (F/T) cycles, of concrete made of 100% FRCA replacement proportioned through various mix proportion techniques (i.e., DRM, EV and PPM), produced with distinct crushing processes (i.e., crusher’s fines (CF) vs. finely ground (FG)) and derived from conventional concrete mixtures incorporating natural and manufactured sand. By using non-destructive techniques (NDT), the compressive strength and the Stiffness Damage Test (SDT), the FRCA concrete mixtures’ inner quality and flaws were appraised before being subjected to F/T cycles as per ASTM C666 [75]. Furthermore, the FRCA concrete mixtures subjected to F/T cycles were assessed using a microscopic technique, the Damage Rating Index (DRI) and its extended version, to better understand the F/T-induced damage development in the recycled concrete mixtures over time.

## 4. Materials and Methods

### 4.1. FRCA Production and Raw Materials Characterization

Two-hundred 35 MPa CC cylinders (100 mm × 200 mm, w/c of 0.47) were fabricated according to ASTM C39 [76] while varying the fine aggregate type (i.e., natural (NS) and manufactured (MS)). A Portland cement (General Use-GU, similar to ASTM Type I) was used along with a coarse limestone aggregate with a nominal maximum size of 19 mm.

Following the CC production, the specimens were de-molded after 24 h and moist cured (100% RH and 20 °C) for 28 days before being processed to produce the FRCA. Figure 2 displays a) the crusher’s fines (CF) and b) fully ground (FG) crushing procedures used to produce the FRCA. For the CF–FRCA, concrete specimens were jaw crushed twice on a 19 mm maximum gap opening then sieved to divide the particles sizes; only the fine fraction of the crushed material was used. Meanwhile, the FG-FRCA was produced by crushing the CRCA material obtained from sieving greater than 4.75 mm through multiple series of crushing (5 mm gap opening in the jaw crusher). Figure 3 shows the particle size distribution of all fine aggregate used in this study, including the specified limits as per [4]. 

Further characterization of the residual cement paste (RCP) content was performed for each type of FRCA through the soluble silica sub-procedure according to ASTM C1084 [77] and C114 [78]. This procedure, as detailed in [46], was selected due to the use of coarse limestone aggregate thus, preventing the selection of other chemical procedures such as maleic acid digestion. Moreover, specific gravity and water absorption (Table 1) were evaluated according to the new method proposed by Rodrigues et al. [79], in which FRCA samples are first saturated in a 0.1% dispersant solution to prevent FRCA cohesion before continuing with a similar procedure as per [4].

In order to produce the FRCA concrete mixtures, the same General Use-GU Portland cement and coarse limestone aggregate (19 mm of maximum nominal size) were selected for use. Their physical properties were determined as shown in Table 2. Moreover, a limestone filler with a particle size distribution (Figure 4) smaller than the Portland cement (i.e., performance filler) was added to the mixtures proportioned through the PPM method to enhance its eco-efficiency; this is further discussed in the following section.

### 4.2. Mix Design Procedures and Proportions

Three distinct FRCA concrete mixtures proportioned with distinct techniques (i.e., DRM, EV and PPM) and containing 100% replacement of the fine aggregate by FRCA were designed (Table 3). Moreover, two control mixtures (e.g., 0% FRCA) incorporating natural (NS) and manufactured (MS) sand were also mix-proportioned for comparative purposes. The water-to-cement (w/c) ratio of all FRCA mixtures was selected as 0.35. In addition, an air-entraining agent (AEA) was used in all mixtures (i.e., control and recycled) to incorporate an amount of air of 7–8% according to [4], as required for structural concrete exposed to freeze-thaw cycles. Lastly, a combination of polycarboxylate-based high range and mid-range water reducer was used in all mixtures at different amounts to achieve consistency (i.e., slump value) of 100 ± 20 mm. Note that the EV and PPM methods account for the RCP attached to the FRCA particles; thus, the mixtures proportioned by these procedures presented much lower Portland cement content. 

### 4.3. FRCA Concrete Manufacturing 

Twenty-seven cylindrical (100 mm × 200 mm) specimens were produced for the CC and FRCA concrete mixtures. Likewise, three 75 mm × 75 mm × 300 mm prisms per concrete mixture (recycled or conventional) were fabricated for the accelerated freeze-thaw tests as per ASTM C666 [75]. All specimens were fabricated according to CSA A23.2-3C [4], de-molded after 24 h and moist-cured (100% RH and 20 °C) for 28 days before testing. 

### 4.4. Hardened Properties

Non-destructive techniques (i.e., electrical resistivity (ER) and dynamic modulus of elasticity through ultrasonic pulse velocity test (UPV)) and compressive strength tests were performed on three specimens from each concrete mixture at 28 days. Furthermore, the stress–strain relationship of selected mixtures before being subjected to F/T was evaluated at 28 days through the use of the Stiffness Damage Test (SDT) as per Sanchez et al. [61,62]. For the SDT, concrete cylinders were subjected to five compressive loading–unloading cycles at a controlled loading rate of 0.10 MPa/s up to 40% of the 28-day compressive strength. 

### 4.5. Resistance to Freeze-Thaw Cycles

All recycled mixtures fabricated were tested against F/T cycles through the use of the accelerated laboratory test procedure as per ASTM C666 [75]. The procedure consists of storing the prisms in an F/T chamber, with temperatures ranging from −18 to 4 °C at time intervals not less than two nor more than five hours and producing a total of 300 cycles. Before beginning the F/T cycles, the mass, average length and fundamental transverse frequency as per [80] were recorded. The specimens were removed from the F/T chamber every 36 cycles of exposure (approximately one week) for further mass, length and frequency evaluations and were continuously monitored over time until 300 cycles were reached. 

The relative dynamic modulus of elasticity of each sample was calculated based on the fundamental transverse frequency measurements, as shown in Equation (2):(2)$$P_{c}=\frac{n_{1}{}^{2}}{n^{2}}\times 100$$
where the percental relative dynamic modulus of elasticity after “c” cycles of F/T is represented by *P_c_*; *n* is the fundamental transverse frequency (Hz) at 0 cycles and *n*_1_ is the fundamental frequency after “c” cycles of F/T. It is worth noting that the test was performed up until 300 cycles or until the sample reached a 40% decrease in the initial relative dynamic modulus of elasticity value.

The length changes were measured according to ASTM C490 [81] and calculated as per Equation (3):(3)$$L_{c}=\frac{l_{2}-l_{1}}{L_{g}}\times 100$$
where *L_c_* (%) is the length change after “c” cycles of F/T; *l*_1_ and *l*_2_ are the comparator length readings at 0 and “c” cycles, respectively, and *L_g_* is the effective gauge inner length measured between the gauge studs. As per ASTM C 666 [75], an expansion of less than 0.10% is the limit distinguishing F/T resistance. 

Finally, the overall performance of the mixtures against F/T action was evaluated based on the durability factor (*DF*) shown in Equation (4):(4)$$DF=\frac{P_{N}N}{M}$$
where *DF* is the durability factor of the specimen (0–100%), which corresponds to the reduction in dynamic ME over freeze-thaw cycles; *P_N_* (%) the dynamic ME after the Nth F/T cycle; *N* refers to the number of cycles at which the dynamic ME reaches the specified minimum value before discontinuing the test or the maximum number of cycles for the analysis (the lower value); and *M* is the specified number of cycles at which the exposure is to be terminated (i.e., 300 cycles).

### 4.6. Microscopic Assessment

The Damage Rating Index (DRI) was used to assess the F/T damage in FRCA concrete mixtures. Concrete prisms subjected to 300 cycles of F/T were sawn in half longitudinally using a masonry saw equipped with a diamond blade followed by successive grinding and polishing (i.e., 60, 140, 280, 600, 1200 and 3000 grits) until a flat reflective surface was achieved. A grid of 1 cm × 1 cm squares was drawn on this surface in which distinct types of cracks (i.e., closed crack in the aggregate (CCA), open crack in the aggregate (OCA), cracks in the residual and new cement paste (CCP, RCP and CCP, NCP, respectively)) were counted using a stereomicroscope at 16× magnification. Those cracks were then weighted using the weighting factors proposed by [71] and normalized to 100 cm^2^ for comparative purposes. Moreover, the extended version of the DRI accounts for the cracks without any weighting factors in absolute and relative counts [66,68].

## 5. Results

### 5.1. Non-Destructive Techniques (NDT), Hardened Properties and Stress–Strain Relationship

The inner quality of the concrete before being subjected to freezing and thawing (F/T) cycles was evaluated through non-destructive techniques (NDT), hardened state properties and the stress–strain relationship. Figure 5a–c display the average surface ER, compressive strength, dynamic modulus elasticity (obtained from the UPV) as well as the static modulus of elasticity (acquired from the stiffness damage test—SDT), respectively, for the FRCA concrete mixtures. Generally, the DRM-proportioned mixtures show the lowest results, followed by the EV-proportioned mixtures, with the PPM-proportioned mixture showing the highest results. The highest surface ER result obtained from the PPM-proportioned mixtures ranged from 10.8 to 14.3 kΩ∙cm with a standard deviation of 3.5 kΩ∙cm for the CF–FRCA and 5.1 kΩ∙cm for the FG-FRCA, while the EV-proportioned mixtures yielded values ranging from 6.5 to 7.3 kΩ∙cm. The DRM-proportioned mixtures show the lowest ER (3.9 to 4.4 kΩ∙cm). Similarly, the dynamic moduli of elasticity of 42 to 51 GPa, 27 to 35 GPa and 15 to 16 GPa were obtained for the PPM, EV and DRM-proportioned mixtures, respectively, while 49 GPa was achieved for both CC mixtures (i.e., natural sand (NS) and manufactured sand (MS)). Likewise, targeting a compressive strength of 35 MPa, values ranging from 65 to 41 MPa (standard deviation of 0.85 MPa for the CF–FRCA and 2.13 MPa for the FG–FRCA) were achieved with PPM-proportioned FRCA mixtures whereas 31 to 26 MPa (overall standard deviation of 1.02 MPa) were achieved for the EV-proportioned mixtures followed by the DRM-proportioned mixtures with an average of 20 MPa (overall standard deviation of 0.8 MPa). The influence of the aggregate type (i.e., NS vs. MS) within the FRCA was not captured through the ER, dynamic modulus of elasticity and compressive strength; however, the results were indeed influenced by the crushing procedure especially captured by the PPM-proportioned mixtures, showing higher results for the fully ground (FG)–FRCA mixtures compared to those of the crusher’s fines (CF)–FRCA.

The influence of the different mix designs used to proportion the FRCA mixtures was also evaluated using the SDT before being subjected to F/T to appraise the inner quality and flaws (i.e., porosity and presence of microcracks within the FRCA) of the recycled mixtures. The SDI, an output parameter obtained through the SDT, represents the amount of energy required to close inner flaws, while the PDI represents the plastic deformation of concrete subjected to loading. Due to testing limitations, only the natural sand (NS) was selected for evaluation along with the CF crushing procedure since the compressive strengths for PPM-proportioned mixtures made with CF–FRCA were closer to the targeted 35 MPa (i.e., 26 MPa and 41–42 MPa, respectively) compared to the FG–FRCA. Moreover, the PPM-proportioned FRCA concrete mixtures were compared by the crushing procedure due to the significant difference observed through the NDT and compressive strength results. The SDT appraises three important parameters SDI, PDI and static modulus of elasticity, which are presented in Table 4. Interestingly, the DRM-proportioned FRCA concrete yields an SDI of 0.10 (standard deviation of 0.02), similar to the SDI obtained for the PPM-proportioned mixtures (i.e., 0.11 with a standard deviation of 0.002 for the CF–FRCA and 0.10 with a standard deviation of 0.01 for the FG-FRCA). The EV-proportioned mixture, on the other hand, shows an SDI of 0.20 (standard deviation of 0.04). The same trend is generally followed for the PDI and static modulus of elasticity, where the EV-proportioned mixture presents the poorest performance with a PDI of 0.16 (standard deviation of 0.05) and a static modulus of elasticity of 21 GPa (standard deviation of 4.1). Meanwhile, the DRM-proportioned mixture presents a better performance with a PDI of 0.09 (standard deviation of 0.01) and a static modulus of elasticity of 26 GPa, (standard deviation of 3.6), yet the PPM-proportioned mixture presents a PDI of 0.07 and 0.08 (standard deviation of 0.001 and 0.02, respectively) for the CF and FG crushing procedure, respectively. The static elastic modulus of the PPM-proportioned mixtures was found to be 29 GPa and 38 GPa (standard deviation of 0.13 and 0.21, respectively) for the CF and FG crushing procedure, respectively. In addition, the SDI, PDI and static modulus of elasticity of CC were found to be 0.12, 0.11 and 45 GPa, respectively, representing the values obtained for a sound concrete.

### 5.2. Freeze-Thaw (F/T) Resistance

#### 5.2.1. Mass Losses

After being subjected to the F/T cycles, the mass loss of the concrete specimens was measured. Figure 6 presents the average mass loss (i.e., residual mass) results for all FRCA mixtures as a function of the number of cycles showing a decrease in the mass as the number of cycles increases. The PPM-proportioned mixtures have the best performance against F/T damage with less than 20% mass loss after 300 cycles (standard deviation ranging from 0.23 to 0.31% throughout the cycles), whereas the EV and DRM-proportioned mixtures show mass losses of up to 45% (standard deviation ranging from 0.08 to 0.24% and 0.17 to 0.39%, respectively, throughout the cycles). Although similar, the DRM-proportioned mixtures following the 300 cycles appeared the most damaged (Figure 7), where excessive spalling on the surfaces along with rounded-edges were observed compared to the EV-proportioned mixtures, with the PPM-proportioned mixtures showing the least amount of surface damage. The DRM-proportioned FG-FRCA mixture incorporating NS did not attain 300 cycles, losing more than 40% of its initial mass at 150 cycles. 

#### 5.2.2. Length Changes

The average changes in length for all FRCA concrete mixtures subjected to F/T cycles are displayed in Figure 8, including the 0.10% limit proposed by ASTM C666 [75]. All DRM-proportioned mixtures exceed the proposed limit at the end of the 300 cycles (standard deviation ranging from 0.07 to 0.35% throughout all cycles), while both EV- and PPM-proportioned mixtures remain below this limit at 0.10% (standard deviation ranging from 0.01 to 0.04%) therefore, deemed resistant to F/T. 

#### 5.2.3. Residual Dynamic Modulus of Elasticity

The average residual dynamic modulus of elasticity results for all FRCA mixtures are illustrated in Figure 9, including the ASTM C666 [75] limit considering concrete to be resistant to F/T when the reduction in dynamic modulus of elasticity should be below 40% at 300 cycles. It may be observed that, once again, the DRM mixes yielded the lowest results among the different mixtures, while PPM mixtures showed the best performance (standard deviation ranging from 12 to 35.5% and 4.9 to 21.6%, respectively, throughout the cycles). The EV mixtures also demonstrated acceptable behavior (i.e., about 38% damage at 300 cycles with a standard deviation ranging from 7.3 to 23.4%). Finally, no significant trend was observed for different types of FRCA (i.e., crusher’s fines/fully ground (CF/FG) and manufactured sand/natural sand (MS/NS)); hence, the mix design procedure poses a greater influence on the F/T resistance of FRCA.

#### 5.2.4. Durability Factor

The durability factors for each FRCA mixture were calculated following the F/T cycles using Equation (4) and are presented in Figure 10. By analyzing the results, one sees that PPM-proportioned mixtures represent the best performance, with all mixtures having a durability factor above 70%, followed by EV and DRM-proportioned mixtures with above 60% and from 20 to 55%, respectively. No apparent trend was observed for the different types of original fine aggregate used (i.e., NS vs. MS) and the crushing procedure (i.e., CF or FG). 

### 5.3. Damage Rating Index (DRI)

The Damage Rating Index (DRI) was used to complement the above results obtained and better understand the cracking behavior of recycled mixtures using FRCA. Figure 11 displays the cracking behavior of concrete mixtures made of FRCA (i.e., MS-FRCA) subjected to 300 cycles of F/T through the DRI bar chart (Figure 11a), the counts per square centimeters (Figure 11b) and proportions (Figure 11c) as per the extended version of the DRI. In general, the FRCA concrete mixtures follow the same trends as previously discussed, where the PPM-proportioned mixtures show a higher resistance to cracking while the DRM-proportioned mixture produced the most cracks after the 300 cycles. Interestingly, the EV-proportioned mixtures showed less damage through the DRI and its extended version, although its F/T performance (i.e., mass loss and residual dynamic modulus of elasticity) was similar to that of the DRM-proportioned mixture highlighting the enhancement of the FRCA concrete when using EV as a mix design method. 

Overall, more cracks are observed in the new cement paste (i.e., CCP, NCP) for all FRCA concrete mixtures, while very few cracks in the residual cement paste (i.e., CCP, RCP) were observed. Although the total number of cracks in the DRM-proportioned mixture (i.e., DRI number of 1090 and 693 counts/100 cm^2^) is higher than other mixtures (i.e., DRI numbers and counts cm^2^ ranging from 370 to 883 and from 267 to 436, respectively), its number of cracks in the cement paste and its proportions (i.e., 232 counts/cm^2^ and 35%, respectively) are lower than the EV-proportioned mixtures, which range from 236 to 257 counts/cm^2^ and 60 to 70% for the crusher’s fines (CF) and finely ground (FG) FRCA, respectively. Otherwise, a lower number of cracks in the cement paste is observed in the PPM-proportioned mixtures at 91 and 231 counts/cm^2^ for the CF and FG FRCA, respectively.

## 6. Discussion

### 6.1. Effect of Materials and Mix Design on FRCA Concrete 

Several mixture design procedures were used to proportion recycled concrete made of 100% FRCA replacement incorporating two different types of original fine aggregates (i.e., natural sand and manufactured sand) and crushed using two distinct processes (i.e., crusher’s fines and fully ground) to evaluate their resistance to freezing and thawing further. 

The electrical resistivity and dynamic modulus of elasticity (i.e., NDTs) provide insight into the inner quality (i.e., porosity) and flaws (i.e., presence of microcracks within the FRCA) of the concrete before being subjected to freezing and thawing cycles. As presented in Section 5.1, the DRM-proportioned mixture showed the lowest surface electrical resistivity and dynamic modulus of elasticity (Figure 5a,b, respectively) due to its higher porosity since the residual cement paste is not considered in this mixture design technique, thus increasing the amount of total cement paste in the concrete. Conversely, EV and PPM mixtures were designed to account for the residual cement paste, increasing the inner quality as described by the surface electrical resistivity and dynamic modulus of elasticity. However, when the PPM mix design is used, the mixtures’ porosity is further reduced due to the enhancement of the particle distribution [51]. Moreover, the type of original fine aggregate (i.e., natural and manufactured sand) within the FRCA was not captured through the surface electrical resistivity; however, the FRCA made of manufactured sand provided slightly higher results when compared to FRCA made of natural sand through the dynamic modulus of elasticity for the EV- and PPM-proportioned mixtures. Although the EV- and PPM-proportioned mixtures account for the residual cement paste adhered to the original aggregate particles, the crushing procedure influences the inner quality of the FRCA particles and consequently of the concrete. The fully ground crushing process exposes more original aggregate (i.e., residual sand and fragments of original coarse aggregate) due to its excessive processing. Therefore, a lower residual cement paste content will result in FRCA particles having more original aggregates exposed at the surface of the FRCA (Figure 12), which in turn reduces the number of interfaces formed within the FRCA concrete and reduces its porosity. 

The same trend is observed through the compressive strength; however, the influence of the crushing procedure is apparent for the PPM-proportioned mixtures. The fully ground FRCA produced significantly higher results at 62 MPa and 65 MPa for the natural and manufactured sand, respectively, although a 35 MPa concrete was targeted, yet not achieved, for the mixtures proportioned through the EV and DRM (i.e., 26–31 MPa and 19–21 MPa, respectively). Nevertheless, PPM-proportioned FRCA mixtures, especially the fully ground FRCA, present better results when compared to EV- and DRM-proportioned mixtures, thus producing FRCA concrete with a better overall inner quality (i.e., porosity). Besides the lower amount of residual cement paste of the fully ground FRCA, the improvement of the FRCA–PPM–FG mixture may also be explained due to the content of the inert filler. Since the volume of filler added in PPM mixtures was equivalent to the volume of the residual cement paste of FRCA, FRCA–PPM–FG mixtures resulted in higher cement content and lower filler content than FRCA–PPM–CF mixtures. This slightly higher cement content might have improved the bond between the FRCA particles, and the new cement paste is fully ground mixtures due to the natural self-healing mechanism developed by the hydration of Portland cement, a phenomenon widely reported in the literature. 

### 6.2. Mechanical Response of FRCA Concrete Subjected to Cyclical Loading

Despite the results obtained through non-destructive techniques (NDTs) and the compressive strength, the EV-proportioned mixtures displayed the lowest performance (i.e., SDI, PDI and modulus of elasticity) out of all mixture proportions used for FRCA concrete. As such, the SDIs of the DRM-proportioned mixtures were found to be similar to those of the PPM-proportioned mixtures, both of which resemble sound concrete. However, the same constituents were used in all mixtures, yet at different proportions, which were captured by the SDT through the SDI and PDI (Table 4). Although the SDI and PDI indicate the inner quality of the concrete mixtures without the influence of their strength differences and corresponding loading, the slopes of the curves and hysteresis area (i.e., area under the stress–strain curve) obtained through cyclic loading may provide additional insight. Figure 13 shows the stress–strain relationship under cyclic loading for each mix design incorporating natural sand and crusher’s fines FRCA including the fully ground FRCA for the PPM mix design due to its apparent difference observed in the NDTs and compressive strength results. The CC (i.e., ACI–NS) represents sound concrete. The slopes of the PPM-proportioned mixtures are steeper compared to the DRM and EV mixtures, therefore, representing a higher stiffness; however, the CC displayed the steepest slope. The EV-proportioned mixture presents the greatest energy dissipation (i.e., largest hysteresis area) and plastic deformation followed by the PPM then the DRM proportioned mixtures. Interestingly, the initial loading slope of the PPM mixture incorporating the fully ground FRCA is similar to that of the CC. Moreover, the fully ground FRCA presents a steeper slope, thus stiffer than that of the crusher’s fines FRCA, which was not evident through the SDI, PDI and modulus of elasticity obtained through the SDT. 

Since the residual cement paste was accounted for in EV-proportioned mixtures and disregarded in DRM, the difference in its performance may be justified by the higher FRCA content present in EV mixtures (714–752 kg/m^3^) when compared to DRM mixtures (524–551 kg/m^3^). Hence, the overall stiffness of the concrete may be governed by the higher number of interfaces found within the FRCA concrete. On the other hand, the PPM-proportioned mixtures contain the most FRCA (879–915 kg/m^3^), thus comprised of an even higher number of interfaces; however, all particles were more efficiently distributed/packed in a system with lower porosity. 

### 6.3. Damage Propagation in FRCA Concrete after 300 Freezing and Thawing Cycles

The resistance to freezing and thawing cycles of concrete is crucial in harsh environments such as those experienced in Canada. The freezing and thawing resistance was therefore appraised for each type of mixture proportioning technique, as well as the nature of the original fine aggregate and crushing procedure. The most apparent difference was between the mixture proportioning techniques while, interestingly, the nature of the original fine aggregate (i.e., natural and manufactured sand) and the crushing procedure (i.e., crusher’s fines and fully ground) did not show any trends. Similar to the non-destructive techniques and compressive strength results, the DRM-proportioned FRCA concrete showed the poorest performance in mass loss, length change and residual modulus of elasticity and exceeded ASTM C666 [75] limits while both PPM and EV-proportioned mixtures conformed to the standard (Figure 8 and Figure 9, respectively). The PPM-proportioned mixtures displayed the highest resistance to freezing and thawing, while the EV-proportioned mixtures were only slightly above the residual modulus of elasticity requirements, although it did not record any change in length and its mass loss was similar to that of the DRM-proportioned mixtures. 

In order to obtain a more thorough understanding of the resistance to freezing and thawing of the various concrete mixtures made of FRCA, a thorough microscopic analysis through the Damage Rating Index (DRI) method was performed. Consequently, the DRM-proportioned mixture evaluated in the microscopic analysis showed a significant number of open cracks within the aggregates, usually associated with distress as opposed to closed cracks in the aggregates (Figure 11), and propagating into the new cement paste (Figure 14a). Meanwhile, the cracks tend to propagate through the bulk, new cement paste for FRCA concrete mixtures proportioned with EV and PPM and do not propagate through the FRCA particles; however, they may propagate in the interfaces formed between the FRCA particles and new mortar (Figure 14b). 

Noticeably, a similar trend is observed for the PPM- and EV-proportioned mixtures where fully ground FRCA was more susceptible to cracking than the crusher’s fines FRCA. Before being subjected to freezing and thawing, the fully ground FRCA exhibited better results than the crusher’s fines. A fully ground FRCA is subjected to a more intensive crushing procedure, hence the higher removal of residual cement paste, exposure of residual sand and original virgin aggregate (OVA) and potentially the introduction of a higher number of microcracks; however, cracks produced by freezing and thawing were uncommonly observed within the FRCA and mostly present within the new cement paste. The crushing procedure, therefore, influenced the cracking behavior of the FRCA concrete such that the formation of new interfaces may govern this behavior where residual to new mortar interfaces show more resistance to cracking than residual sand/OVA to new mortar interfaces due to the difference between the stiffness of the components [82]. The literature suggests that the strength of concrete made of RCA is governed by its weakest ITZ [83,84,85], while a stronger bond is observed between the residual to new mortar interface [64,83,86,87]. In addition, more FRCA was used in mixtures containing the fully ground FRCA, thus creating more interfaces and weaker zones within the concrete. Similarly, the PPM-proportioned mixtures contain higher amounts of FRCA and lower coarse aggregate content when compared to the EV-proportioned mixtures, thus reducing the differential stresses within the recycled concrete proportioned through PPM and, therefore, the number of cracks. Moreover, PPM–FG mixtures contain lower residual cement paste and filler content, which may also contribute to the higher number of cracks in the new cement paste when compared to PPM mixtures made with crusher’s fines FRCA. The additional content of fillers in the PPM mixtures made with crusher’s fines FRCA may also have contributed to the quality of its new interface. Nevertheless, crusher’s fines FRCA are a by-product of coarse RCA production, which require less processing, hence energy to produce compared to the fully ground FRCA.

## 7. Conclusions

Three types of mix proportioning techniques (DRM, EV and PPM) were used in this work and assessed by evaluating the inner quality and the resistance to freezing and thawing of FRCA concrete designed with 100% replacement. The main findings can be highlighted hereafter:The PPM-proportioned mixtures showed the highest inner quality before being subjected to freezing and thawing, followed by the EV and DRM-proportioned mixtures. The influence of the type of residual sand was not as significant as the crushing procedure, indicating that FRCA subjected to a more rigorous crushing sequence presented a better inner quality;The EV and PPM mixture proportioning techniques produced a concrete made of FRCA having adequate freezing and thawing resistance, while the DRM-proportioned mixtures were not considered resistant to freezing and thawing. The PPM mixtures showed the best performance followed by EV then DRM-proportioned FRCA concrete mixtures while no apparent trend was observed between the type of residual sand and crushing procedure;The overall durability factor of all FRCA mixtures subjected to freeze-thaw cycles was observed to have considerable variation between the mix design methods. A higher durability factor was observed for PPM mixtures, followed by EV then DRM. This demonstrates that the mix design procedure adopted to design FRCA concrete is more important than the material’s quality; PPM and EV-proportioning techniques being capable of reducing the variability presented while using the DRM;The DRI captured the differences in the damage propagation of FRCA concrete subjected to freezing and thawing. The highest level of damage was observed in DRM-proportioned mixtures, whereas the EV- and PPM-proportioned mixtures showed less damage. Despite showing a better performance before being subjected to freezing and thawing, the fully ground FRCA concrete presented more damage compared to the crusher’s fines FRCA. The crushing procedure significantly influences the mechanical properties, inner quality and crack generation and propagation in FRCA concrete. Further research is therefore required to understand the cracking behavior of RCA (i.e., coarse and fine) concerning its multi-phase nature.

The freezing and thawing resistance of concrete made of FRCA was significantly improved when using mixture proportioning techniques different than the direct replacement method, which failed to comply with the specified requirements. Although it is the most used mix design technique in practice, it is evident that direct replacement methods not only show the poorest performance overall but also require a much higher cement content resulting in an unsustainable approach. Furthermore, it is clear that the inner quality of FRCA manufactured through multiple series of crushing (i.e., FG) is enhanced when compared to only two series (i.e., CF); However, when the proportion was properly mixed, recycled concrete mixtures made of both FG and CF particles displayed quite suitable results. Therefore, it is recommended that the CF method be adopted since it requires less energy and thus provides a more sustainable material with less generation of waste.

## Figures and Tables

**Figure 1 materials-15-01342-f001:**
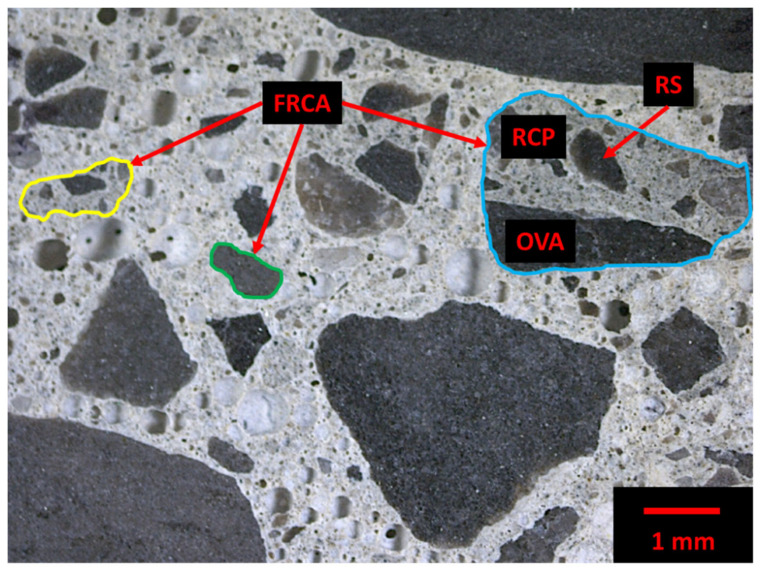
Fine recycled concrete aggregate (FRCA) concrete multi-phase nature including FRCA particles with original virgin aggregate (OVA), residual cement paste (RCP) and residual sand (RS).

**Figure 2 materials-15-01342-f002:**
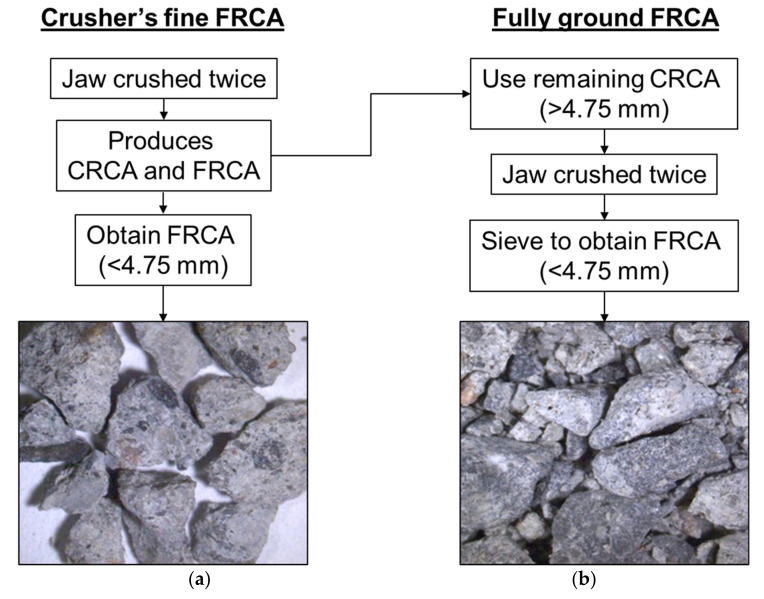
Summary of the two methods of FRCA production adopted in this experimental work (**a**) crusher’s fines (CF) and (**b**) fully ground (FG).

**Figure 3 materials-15-01342-f003:**
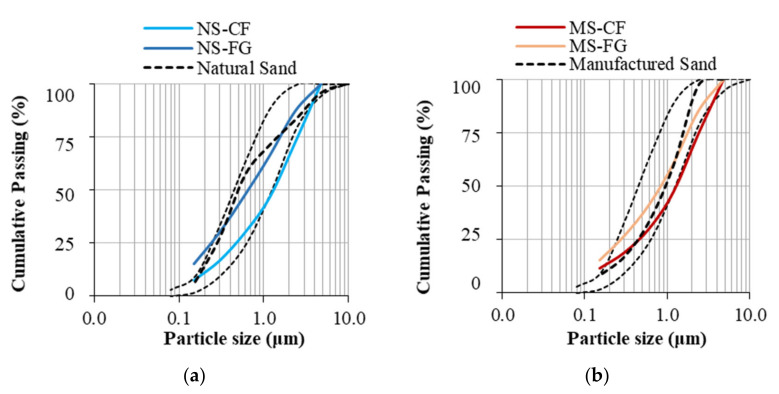
Particle size distribution curves for FRCA derived from (**a**) natural sand (NS) and (**b**) manufactured sand (MS).

**Figure 4 materials-15-01342-f004:**
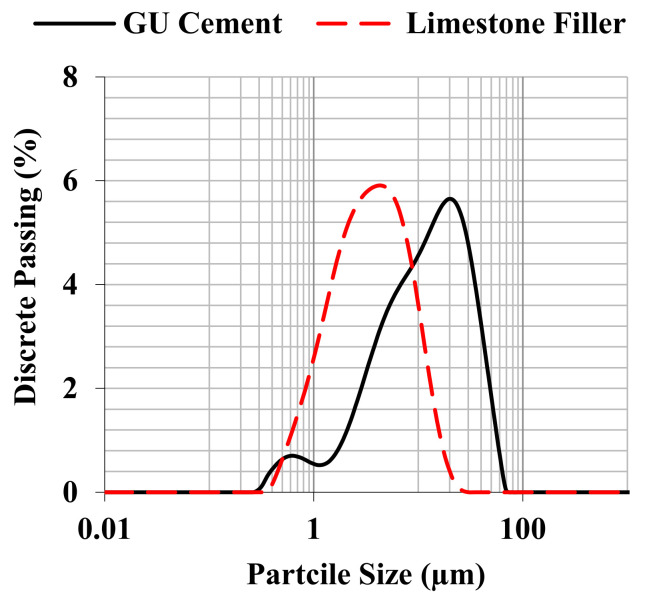
Particle size distribution of Portland cement and limestone filler used in the particle packing model (PPM) mixtures.

**Figure 5 materials-15-01342-f005:**
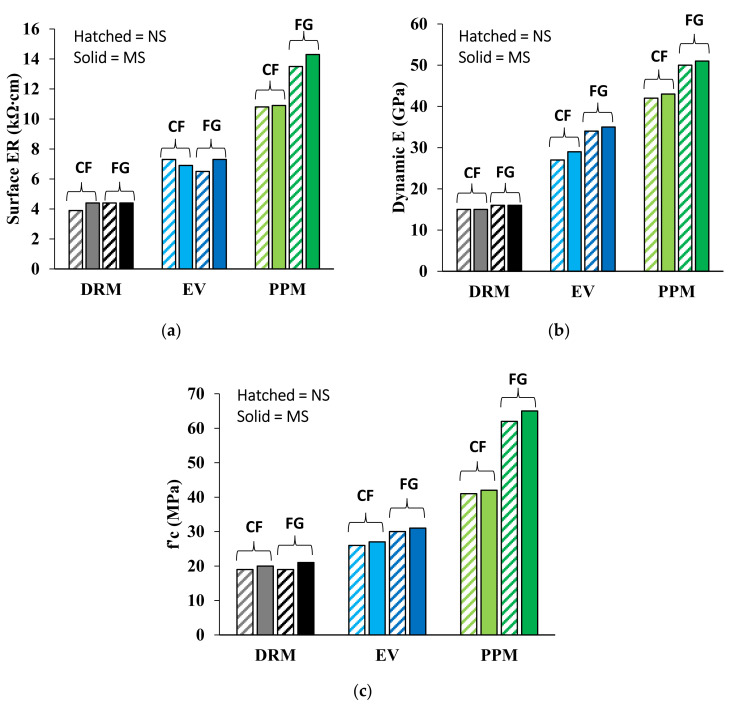
(**a**) Surface electrical resistivity (ER), (**b**) dynamic E and (**c**) compressive strength for all FRCA concrete mixtures.

**Figure 6 materials-15-01342-f006:**
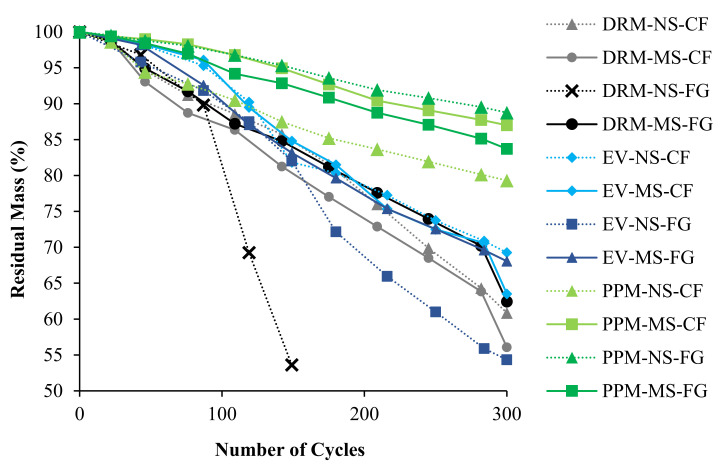
Mass loss as a function of freeze-thaw cycles for all FRCA mixtures.

**Figure 7 materials-15-01342-f007:**
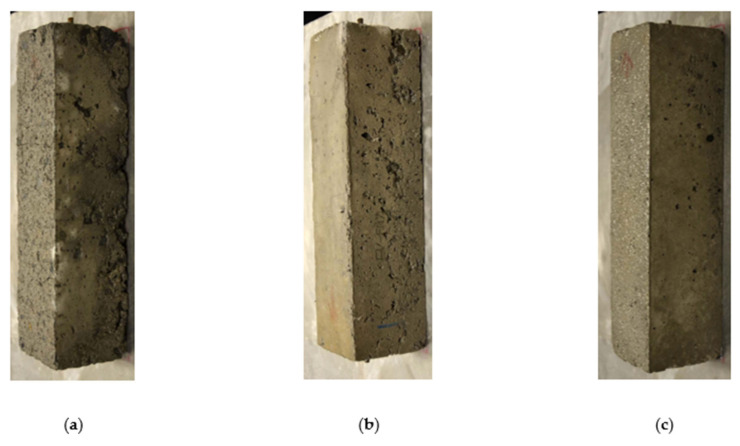
Degradation of FRCA mixes at the end of 300 freezing and thawing cycles using (**a**) direct replacement method (DRM); (**b**) equivalent volume (EV) and (**c**) PPM mixture proportions.

**Figure 8 materials-15-01342-f008:**
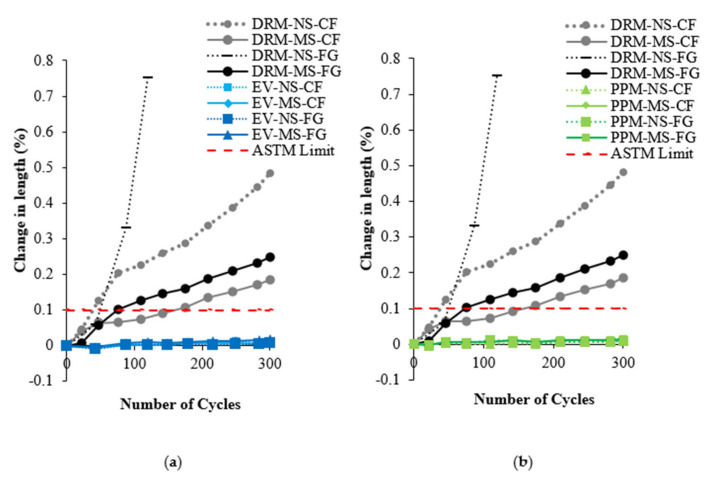
Length change over 300 cycles for (**a**) EV and (**b**) PPM in comparison to DRM mixtures.

**Figure 9 materials-15-01342-f009:**
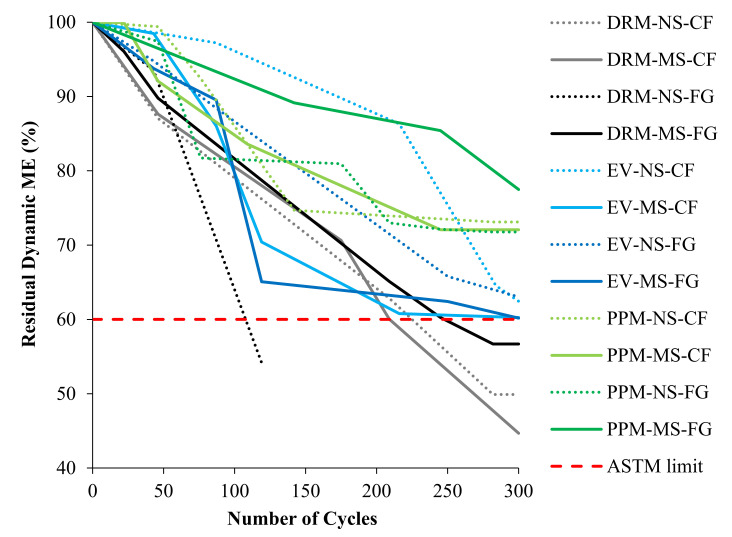
Residual dynamic modulus of elasticity over 300 cycles for PPM and EV in comparison to DRM mixtures.

**Figure 10 materials-15-01342-f010:**
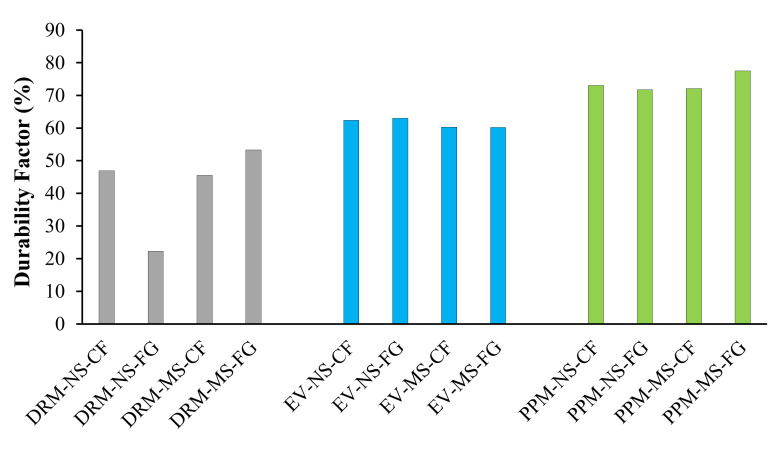
Durability factors of FRCA concrete mixtures subjected to accelerated freezing and thawing cycles.

**Figure 11 materials-15-01342-f011:**
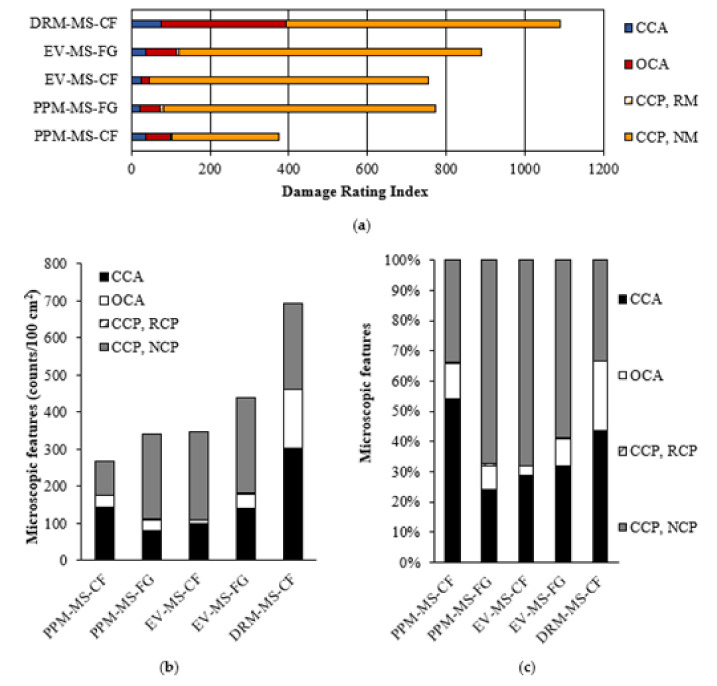
Distress features in FRCA mixtures shown as (**a**) Damage Rating Index (DRI) bar chart, (**b**) counts per 100 cm^2^ and (**c**) percentages.

**Figure 12 materials-15-01342-f012:**
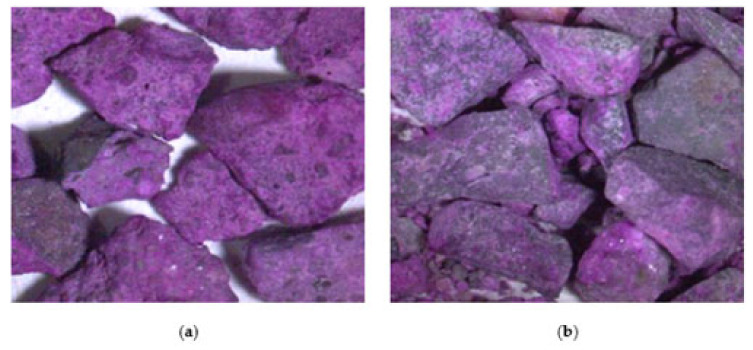
Phenolphthalein application on FRCA produced by (**a**) CF and (**b**) FG.

**Figure 13 materials-15-01342-f013:**
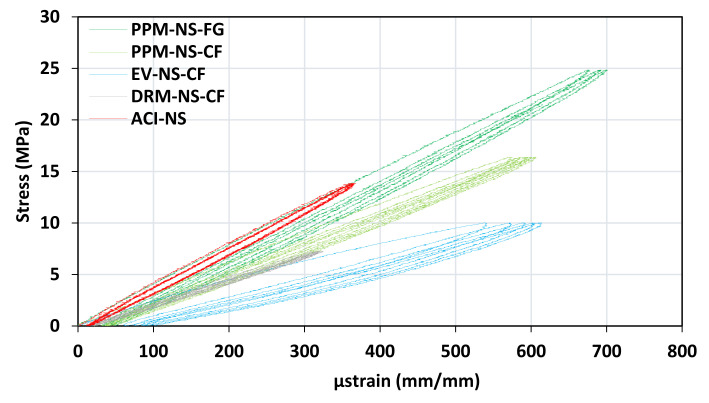
Stress–strain curves obtained from the SDT comparing FRCA–CF and CC mixtures from NS source, and PPM with FRCA of CF and FG types.

**Figure 14 materials-15-01342-f014:**
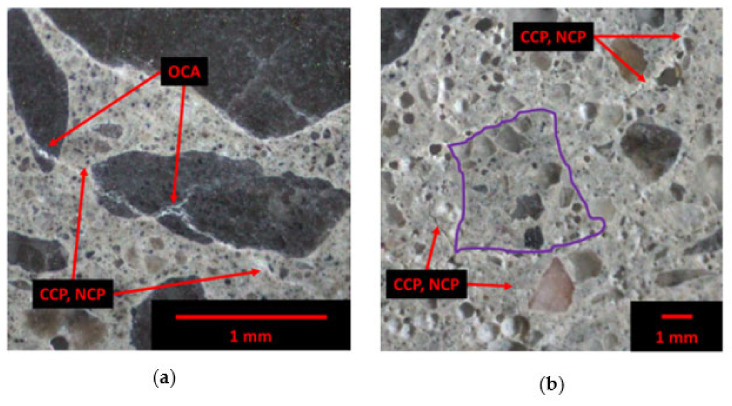
Crack propagation through (**a**) coarse aggregate as open cracks in the aggregate (OCA) and (**b**) in new cement paste (CCP, NCP).

**Table 1 materials-15-01342-t001:** Properties of FRCA and natural aggregates.

Physical Property	FRCA NS–CF	FRCA NS–FG	NS	FRCA MS–CF	FRCA MS–FG	MS	Coarse Limestone
RCP content (wt.%)	15.5	11.5	-	16.8	11.4	-	-
SSD specific gravity (kg/L)	2.47	2.56	2.70	2.51	2.58	2.76	2.79
OD specific gravity (kg/L)	2.32	2.42	2.67	2.36	2.44	2.74	2.78
Water absorption (%)	7.87	6.38	0.86	7.76	6.16	0.65	0.42
Fineness modulus	3.27	2.53	2.59	3.17	2.70	2.85	-

**Table 2 materials-15-01342-t002:** Physical properties characterization.

Material	Mass (g)	Volume (cm^3^)	Specific Gravity (g/cm^3^)	Specific Surface Area (m^2^/g)
Portland cement	31.9	10.49	3.03	1.00
Limestone filler	19.5	7.56	2.60	3.70

**Table 3 materials-15-01342-t003:** FRCA and control concrete mix design proportions.

Mixture	Portland Cement (kg/m^3^)	FRCA (kg/m^3^)	Natural Fine Aggregate (kg/m^3^)	Natural Coarse Aggregate (kg/m^3^)	Limestone Filler (kg/m^3^)	Water (kg/m^3^)	w/c	AEA (%)	Water Reducer (kg/m^3^)
ACI–NS	370	-	738	1032	-	174	0.47	0.65	-
ACI–MS	370	-	759	1032	-	174	0.47	0.65	-
DRM–NS CF	497	524		1032	-	174	0.35	0.45	-
DRM–NS FG	497	546		1032	-	174	0.35		-
DRM–MS CF	497	533		1032	-	174	0.35		-
DRM–MS FG	497	551		1032	-	174	0.35		-
EV–NS CF	374	714	-	1005	-	131	0.35	0.50	1.2
EV–NS FG	373	740	-	1014	-	131	0.35		1.2
EV–MS CF	372	732	-	1004	-	130	0.35		1.2
EV–MS FG	373	752	-	1006	-	131	0.35		1.2
PPM–NS CF	308	879	-	806	108	108	0.35	0.50	1.0
PPM–NS FG	333	907	-	797	83	117	0.35		1.0
PPM–MS CF	299	898	-	809	118	105	0.35		1.2
PPM–MS FG	332	915	-	798	84	116	0.35		1.2

**Table 4 materials-15-01342-t004:** Stiffness damade test (SDT) output.

Mixture	SDI	PDI	Static Modulus of Elasticity (GPa)
DRM–NS–CF	0.10	0.09	26
EV–NS–CF	0.20	0.16	21
PPM–NS–CF	0.11	0.07	29
PPM–NS–FG	0.10	0.08	38

## Data Availability

All the data are available within the manuscript.

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
