# Peer review of "Freezing and Thawing Resistance of Fine Recycled Concrete Aggregate (FRCA) Mixtures Designed with Distinct Techniques"

_materials, 2022, doi:10.3390/ma15041342_

Round 1

Reviewer 1 Report

The paper "Freezing and Thawing Resistance of Fine Recycled Concrete Aggregate (FRCA) Mixtures Designed with Distinct Techniques" has serious flaws in the content, the experiment planning, the structure, and the discussione of the results.

The structure of the paper is not correct; it is not compliant with the template of Materials. It is a messy paper. The reference is not correct. Most of the sentences in the introduction are not referenced.

There are typing errors.

Author Response

Dear Reviewer 1,

First, we would like to thank you for kindly taking the time to review our manuscript. We have taken all of your comments into consideration and have reviewed our manuscript extensively. We have also accounted for all changes requested by the other Reviewers; all changes are highlighted in yellow in the revised version of the manuscript.

Reviewer 2 Report

This study investigated the freeze-thaw resistance of the concrete incorporating FRCA with different mix-proportion techniques. The results indicated that the PPM proportioned mixture showed the best performance and the crushing process presented significant effects on the cracking behavior of the recycled mixtures under Freeze-thaw conditions. The research content is well presented in the manuscript. The research findings facilitate the development of the concrete mixtures with the FRCA and the application of green construction materials. There are only minor issues that need to be addressed:

  1. From lines 141 to 142, 287 to 288, line 295 to 296, and line 303, correct the “Error! Reference source not found”.
  2. The conclusions can be more concise.

Author Response

Dear Reviewer 2, thank you for your positive comments. We greatly appreciate your feedback and time put into the review. We have addressed all the remarks made, which are presented in the attached document.

Reviewer 3 Report

I am really happy to review this manuscript. The present manuscript gives valuable information on the durability of concrete with the recycled aggregate which is a most practical pyproduct materials. The experimental details are well-organized and the analysis of the results is proper and appropriate. I could not find the point to be revised. I recommend to accept this manuscript without any revising.  

Author Response

Dear Reviewer 3, thank you kindly for your comments. We greatly appreciate your feedback and time put into the review.

Reviewer 4 Report

General report and comments:

  • Please rewrite the Abstract section. Half of the description is improper to Abstract. In this section, the authors should give a pertinent overview of the work.
  • Short and laconic. Please extend the review and description of the introduction chapter.
  • The background section consists of only one sentence and one Table. Please delete this table. The abbreviations should be explained in place that first appear in the paper.
  • Line 141, 287, 296, 303. Please verify and correct the reference.
  • Line 148. Please verify the format of the equations, see Materials Template requirements.
  • Line 149. The abbreviations should be like in equation 1, see lack of italic form.
  • Chapter 3 should be moved to the last paragraph of the introduction section. The introduction should define the purpose of the work and its significance. It is a lack of this information in the Introduction chapter.
  • Please rebuild the first three chapters to find the consistent and logical new introduction chapter.
  • Line ASTM C666. The reference to standard should be added.
  • Figure 2. Please correct and improve the quality of the figure.
  • Figure 3. Please verify a lack of reference to this figure in the text. Please add vertical grids.
  • Please verify Materials Template requirements for citation of tables and figures in the manuscript.
  • Figure 5. The mean values are shown. Please add standard error values.
  • Chapter 5. Results. The ANOVA statistical analysis should be performed to find out whether the results of laboratory tests obtained for defined groups of specimens differ significantly from each other.
  • Chapter 5. Most parts begin with the word Figure (see Line 325, 371, 395, 415 etc). Please rebuild and correct.
  • Line 370. The section is beginning with Figure. Please verify.
  • Figure 13. Where are the 13a and 13b figures? Please correct the caption.
  • Line 569-572. Blank lines. Please verify.
  • A greater effort should be done in the conclusion chapter. The authors should directly indicate the exact novelty of this research study after the bullet points. Only laconic conclusions are given. The conclusions should indicate (underlined) directly the new science of this manuscript to the community.
  • There should be closing remarks after the bullet points of conclusions, keeping in mind all the outcomes obtained.

Author Response

Dear Reviewer 4, first we would like to thank you for this extensive review. Your time and efforts are greatly appreciated. You will find the corrections and comments made by the authors in the attaches document.

Reviewer 5 Report

The paper is interesting. Some comments can be found below:

-We live now in a climate emergency so its most strange that the authors have not start the paper by mentioning exactly that. It seems that they are not aware about the words of a Professor of Physics at the University of Oxford authored a paper where one can read the following:

 “Let’s get this on the table right away, without mincing words. With regard to the climate crisis, yes, it’s time to panic”

Pierrehumbert, R., 2019. There is no Plan B for dealing with the climate crisis. Bulletin of the Atomic Scientists, pp.1-7.

So please start the introduction by draw a connection between environmental degradation and resource efficiency.

- “The use of recycled concrete aggregates (RCA) has been progressively considered as a sustainable alternative”

Comment: The environmentak efficiency of RAC depend on the transport distance. See on this issue

Marinković, S.B., Ignjatović, I. and Radonjanin, V., 2013. Life-cycle assessment (LCA) of concrete with recycled aggregates (RAs). In Handbook of Recycled Concrete and Demolition Waste (pp. 569-604). Elsevier

- “ the coarse RCA 40 (CRCA) fraction is used in new concrete construction, often limited to non-structural ap-41 plications due to concerns over the material’s variability, quality, and presence of impu-42 rities (i.e., other waste debris)”

Comment: Provide references

-The paper has too many accronyms. Using acronymous for expressions like “general use”, oven dry, natural sand etc etc make no sense

Author Response

Dear Reviewer 5, first, we would like to thank you kindly for your comments and for taking the time to review our manuscript. We appreciate your interest in our work. The comments made by the authors can be found in the attached document.

Round 2

Reviewer 1 Report

it is not an excellent paper

Reviewer 4 Report

I recommend the paper " Recycled Aggregate Concrete and Alternative Binders for Sustainable Building Engineering” for publication.